# Expectancies of the Effects of Cannabis Use in Individuals with Social Anxiety Disorder (SAD)

**DOI:** 10.3390/brainsci14030246

**Published:** 2024-03-02

**Authors:** Sonja Elsaid, Ruoyu Wang, Stefan Kloiber, Rebecca Haines-Saah, Ahmed N. Hassan, Bernard Le Foll

**Affiliations:** 1Translational Addiction Research Laboratory, Campbell Family Mental Health Research Institute, Centre for Addiction and Mental Health, Toronto, ON M6J 1H4, Canada; sonja.elsaid@camh.ca (S.E.); james.wang@camh.ca (R.W.); ahmed.hassan@camh.ca (A.N.H.); 2Institute of Medical Science, Faculty of Medicine, University of Toronto, Toronto, ON M5S 1A8, Canada; stefan.kloiber@camh.ca; 3Department of Pharmacology and Toxicology, Faculty of Medicine, University of Toronto, Toronto, ON M5S 1A8, Canada; 4Department of Psychiatry, University of Toronto, Toronto, ON M5S 1A8, Canada; 5Campbell Family Mental Health Research Institute, Centre for Addiction and Mental Health, Toronto, ON M6J 1H4, Canada; 6Department of Community Health Sciences, Cumming School of Medicine, University of Calgary, Calgary, AB T2N 1N4, Canada; rebecca.saah@ucalgary.ca; 7Addictions Division, Centre for Addiction and Mental Health, Toronto, ON M6J 1H4, Canada; 8Department of Psychiatry, King Abdulaziz University, Jeddah 22254, Saudi Arabia; 9Department of Family and Community Medicine, Toronto, ON M5G 1V7, Canada; 10Waypoint Research Institute, Waypoint Centre for Mental Health Care, Penetanguishene, ON L9M 1G3, Canada

**Keywords:** social anxiety disorder (SAD), social phobia (SP), cannabis use, cannabis use disorder (CUD), addictions, expectancies of the effects of cannabis use, cannabis use perceptions

## Abstract

Previous research has indicated that anticipating positive effects from cannabis use may be linked with increased frequency of cannabis consumption, yet these expectancies remain poorly understood in adults with social anxiety disorder (SAD). Thus, our study aimed to investigate the expectancies of the effects of cannabis use in 26 frequently using adults with SAD (age: 27.9 ± 7.3 years; 54% female) and 26 (age: 27.4 ± 6.7 years; 50% female) without. While no between-group differences were observed, both groups reported expecting tension reduction and relaxation (F = 0.001; *p* = 0.974), cravings, and physical effects (F = 1.10; *p* = 0.300), but denied global negative effects (F = 0.11; *p* = 0.744). The trajectory of cannabis use perceptions (further investigated in 12/26 participants/group) also showed no between-group differences. Before the initial use, positive perceptions may have led to initial and continuous cannabis consumption, while the symptoms of cannabis use disorder may have contributed to repeated use. Our data indicate that, regardless of psychiatric history, frequent cannabis-using adults are more likely to report positive expectancies, which are often associated with increased patterns of cannabis consumption. Psychoeducational programs and openly discussing the risks of cannabis may be beneficial in preventing and/or reducing cannabis use in people with SAD.

## 1. Introduction

Individuals with social anxiety disorder (SAD) suffer from intense fear, avoidance of social situations, and a wide range of associated problems. Recent evidence shows that the lifetime prevalence of SAD was 12.1% in the US [1,2,3,4,5]. Only 25% of those affected with SAD are estimated to seek treatment, and only 30% are expected to recover [6]. Considering the lack of adequate treatments, some individuals with SAD resort to cannabis use in order to cope with their symptoms [1,7,8,9,10,11,12,13,14,15,16]. For instance, within our research group, 42% of individuals with SAD reported using cannabis as a coping mechanism from their initial use, whereas 100% reported maintaining their use to cope with symptoms of SAD [14]. Furthermore, the data suggest that this patient population is seven times more likely to experience cannabis dependence than the general public [1]. Recently, it was indicated that 29% of individuals with lifetime cannabis dependence had SAD, whereas rates of lifetime SAD were 15.5% among those with cannabis dependence, and 5% among those with cannabis abuse, [1,2,9,17]. Taken together, the research implies that individuals with SAD who use cannabis are at risk of developing cannabis use disorder (CUD) [1,7,8,9,10,11,12,13,14,15,16,17].

To better plan treatment efforts, clinicians have been looking for ways to predict different patterns of cannabis use. One promising area is the expectancy of the effects of drug use theory [18], which also applies to cannabis use. This theory posits that drug-related perceptions may influence drug consumption behavior. Accordingly, expecting positive effects was linked to a greater frequency of cannabis use, whereas expecting adverse effects was negatively correlated [19,20,21].

Previous research on emerging adults with social anxiety (contrary to what was expected) indicated that social anxiety was negatively correlated with expectancies of social and sexual facilitation, relaxation, and tension reduction [9]. Relaxation and tension reduction expectancies mediated the relationship between social anxiety and cannabis craving [22]. The research also showed that individuals with social anxiety did not expect cannabis use to help them cope with negative effects more so than those without symptoms [9]. Moreover, higher levels of social anxiety were positively related to cognitive and behavioral impairment and negative global expectancies (unlike what was predicted by the expectancy of drug use theory) [9,10]. The interpretation of the study findings was that socially anxious cannabis users may have benefited from cannabis-related effects, such as relieving anxious thoughts. Consequently, they may have used cannabis, believing that others would attribute their behavior to cannabis effects, rather than to their personality. These findings aligned with the self-handicapping theory of drug use, indicating that individuals create or claim obstacles (e.g., impairment due to drug use) in anticipation of failed performance (e.g., perceived inadequacies in behavior in social situations) to protect against hurting their own self-esteem [9,10,18,23].

However, most of the abovementioned studies were correlational, focusing on emerging adults residing in the Southern US, who may have had symptoms of SAD, but may not have necessarily been diagnosed with SAD, as per the DSM criteria. Henceforth, it is unclear what expectancies of the effects of cannabis use occur in the general adult population who are diagnosed with moderate-to-severe SAD per the DSM-5. Also, it is still unknown how these expectancies may change before and immediately after the onset of cannabis use, as well as after repeated use. Understanding the time trajectory of the expectancies of cannabis use in adults with SAD would help clinicians better predict their cannabis use patterns, and assist them in developing adequate drug prevention and harm-reduction efforts. Henceforth, the primary objective of this study was to compare the expectancies of the effects of cannabis use between regular cannabis users with and without SAD. Based on previous research in emerging adults, we hypothesized that, compared to their respective controls, adults with SAD will report more expectancies of cognitive and behavioral impairments. The exploratory objective was to examine the differences in cannabis-related perceptions before the use of cannabis, immediately after the initial cannabis use, and several years after repeated use between the two groups, while focusing on how cannabis perceptions and the expectancies of the effects of cannabis use change over time within the SAD group.

## 2. Materials and Methods

The Research Ethics Board of the Centre for Addiction and Mental Health (CAMH), Toronto, ON, Canada (Ethics # 085-2019), originally approved the study on 1 December 2019. The research was conducted under the Declaration of Helsinki, and according to guidelines on research focusing on human subjects. All participants provided written informed consent before enrolment in the study. This study was conducted at CAMH. We employed a convergent mixed-method design in this study [24,25,26].

### 2.1. Participants

Twenty-six participants with and twenty-six without SAD completed the outcome measures component of the study, which included completing questionnaires. Twelve (50% females) with SAD and twelve without SAD were invited on a first come, first served basis to participate in in-depth interviews. The inclusion criteria in the SAD group were (a) males and females, (b) 18 years of age and over, (c) used cannabis at least once per week in the past three months, (d) scored at least 60 total points or above on the Liebowitz Social Anxiety Scale (LSAS) [27]. The participants in this group also had to meet the criteria for SAD according to the Structured Clinical Interview for Diagnostic Statistical Manual Research Version—5 Disorders (SCID-RV for DSM-5) [28]. The exclusion criteria for this group were past or present primary acute or chronic physical/medical and psychiatric illnesses; however, individuals with SAD and comorbid mild major depressive disorder, any other anxiety disorder, cluster C personality disorder, or CUD were included in the study. The inclusion criteria for the control group were (a) males and females, (b) 18 years of age and over, (c) smoked cannabis at least once per week, (d) scored < 30 total points on LSAS, and (e) no past or current psychiatric or medical illnesses (except for CUD). Regular cigarette smokers (those who met the criteria for Nicotine Use Disorder) in both groups were also included in the study.

### 2.2. Study Procedures

Study participants were recruited by posting study flyers at the clinical sites at CAMH, the University of Toronto, in the community, by advertising online using Facebook and Kijiji, and by contacting participants who had previously consented to be informed about the future studies. To all interested parties (329 participants; 267 for the SAD and 62 for the control group), a short pre-screening questionnaire was administered over the phone to provide a summary of the study procedures and conduct a preliminary screening for the inclusion/exclusion criteria. Out of 329, 173 in the SAD group and 30 in the controls did not meet the preliminary criteria, while 27 in the SAD group were not interested in participating in the study. Interested study participants who met the preliminary study criteria (67 in the SAD and 32 in the control group) were invited to participate in the study. Before attending their test day, verbal informed consent was obtained over the phone. Out of the 67 invited participants in the SAD group, 43 consented, while 27 were no-shows at the consenting phone interviews. In the control group, out of the 32 enrolled, 31 consented, whereas only 1 participant did not attend the consenting process. The consent to participate in the study was documented in the informed consent form (ICF), explicitly designed for online participation. Subsequently, participants attended one online WebEx session on the test day. The screening session included administering SCID-5 RV [28] to determine if the participants met the eligibility criteria, followed by the screening checklist for personality disorders, LSAS [27], the Medical and Psychiatric History Form, and the Sociodemographic Assessment Form. In the SAD group, 17/43 did not meet the study criteria during screening, while 5/31 failed screening in the control group. Those who met the eligibility criteria (26 in the SAD and 26 in the control group) participated in the study session, which included administering the Marijuana Expectancy of the Effects Questionnaire [19,20], the Cannabis Use Problems Identification Test (CUPIT) [29], and the Drug/Alcohol Use History Form. The first twelve participants from each group were invited to an in-depth interview on a first come, first served basis. All invited participants accepted the invite to attend in-depth interviews, except one male participant with SAD as he felt uncomfortable answering open-ended questions. Participation in in-depth interviews was optional. All data for this study were collected using paper-based tools.

### 2.3. Outcome Measures

The SCID for the DSM-5 RV [28] is a semi-structured interview, aimed at making a psychiatric diagnosis, according to [30]. Only the main SCID-5-RV modules were administered in this study. The Liebowitz Social Anxiety Scale (LSAS) [27] comprises 24 social situations; each situation is rated on a scale of 0 to 3 for anxiety/fear and avoidance. The LSAS total scores ranged from 0 to 144, according to which, total scores of 60 and above represent a significant degree of social anxiety, whereas <30 scores indicate no presence of social anxiety. The scale itself is normally used to determine the severity of SAD. Previous studies indicated that LSAS exhibited an internal consistency of α = 0.95 and a test–retest reliability of 0.83 [31]. The Medical and Psychiatric History Form is a checklist assessing major physical illnesses and previous clinical diagnoses of mental health conductions. The Sociodemographic Assessment Form indicates the demographic characteristics of the study participants.

The Medical and Psychiatric History Form is a checklist that assesses major physical illnesses and any previous clinical diagnoses of mental health conductions. The Sociodemographic Assessment Form indicates the demographic characteristics of study participants. The Drug/Alcohol Use History Form captured a frequency of lifetime and current (in the past month) alcohol and drug use. Besides cannabis, alcohol, and cigarette use, other types of substance use recorded in this form included sedatives (benzodiazepines, barbiturates, sleeping pills, and quaaludes), stimulants (methamphetamines, amphetamines, ADHD medications, and other diet pills), cocaine, opioids (heroin, morphine, oxycodone, opium, methadone, and Fentanyl), dissociative anesthetics (PCP, ketamine, and GHB), and hallucinogens (LSD, mushrooms, MDMA, and peyote).

The Marijuana Expectancies of the Effects Questionnaire (MEEQ) is a 48-item survey regarding the expectancies related to cannabis use [19,20]. The questionnaire consists of six lower-order domains, which demonstrated adequate internal consistency as follows: Cognitive/Behavioral Impairment (α = 0.87), Relaxation and Tension Reduction (α = 0.90), Social/Sexual Facilitation (α = 0.70), Perceptual/Cognitive Enhancement (α = 0.85), Global Negative Effects (α = 0.87), and Craving/Physical Effects (α = 0.90). Relaxation and Tension Reduction, Social and Sexual Facilitation, and Perceptual Cognitive Enhancement are considered positive, whereas Global Negative Effects and Cognitive and Behavioral Enhancements are negative expectancy domain. Craving and Physical Effects is considered a neutral domain. Three positive domains are included in the higher-order Positive Expectancies with (α = 0.70), whereas the two negative domains are part of the higher-order Negative Expectancies (α = 0.91) domain. Participants rate each question on a scale of 1–5 (1, strongly disagree; 2, somewhat disagree; 3, neutral; 4, somewhat agree; and 5, strongly agree) [19,20].

The Cannabis Use Problem Identifications Test (CUPIT) [29] identifies problematic cannabis use. The total CUPIT scores of 20 or above indicate the occurrence of CUD. Scores 12–20 suggest that CUD may occur soon. CUPIT exhibited good-to-excellent test–retest (0.89–0.99) reliability and internal consistency (0.92–0.83) [29].

### 2.4. In-Depth Interviews

The in-depth interviews aimed to further explore the cannabis use perceptions beyond the expectancies of the effects of cannabis use described in the MEEQ. The questionnaire focuses on exploring cannabis use perceptions in both groups at the three time points. During the interview, participants were asked to recall their expectancies of the effects of cannabis use and other perceptions before the initial use, immediately after the initial use, and approximately 8–11 years after repeated use.

### 2.5. Data Analysis

Considering that previous research mainly focused on emerging adults and not on the general adult populations, our sample size calculations were conducted on the data presented in [8] from emerging adults, aged between 18–22 years of age (mean = 19.13 years of age, standard deviation = 1.07) [8]. G*Power 3.1 (Heinrich Heine University of Dusseldorf, Germany) was used for computing the effects and sample size calculations based on the means and standard deviations of the MEEQ Cognitive and Behavioral Impairments domain (hypothesis), used from SAD and non-SAD participants [8]. The derived effect size was Cohen’s d = 0.82, which was entered with 1 − β = 0.8 and α = 0.05 to determine that 26 participants per group were a sufficient sample size for our quantitative analysis.

Statistical analysis was performed using the SPSS software, version 28.0 (IBM Company, Armonk, NY, USA) for the quantitative and qualitative data. For the comparison of parametric variables, an independent *t*-test was used to compare the means. MEEQ scores for each domain were compared using a univariate ANOVA test. Linear regression was conducted to assess the contribution of group, sex, age, and current CUD to the dependent variables (each MEEQ domain), and to assess the contribution of predictor variables that included group, sex, age, and marijuana effects expectancies (each MEEQ domain separately) to the problematic cannabis use (CUPIT), which was, in this analysis, entered as a dependent variable. A Chi-square test and odds ratios were used to compare the non-parametric data. The Pearson’s correlation test examined the direct correlation between demographic and clinical variables for the SAD group. The significance of the data was set at *p* ≤ 0.05.

The recorded in-depth interviews were transcribed verbatim and imported into NVivo software (NVivo 12 Plus, QSR International, Burlington, MA, USA). The detailed theme analysis [24,25,26] was subsequently conducted by authors (SE and ANH) independently, followed by organizing agreed-upon codes into distinct themes in NVivo. The overlapping themes were merged into one theme after reaching a consensus between the two authors [24,25,26]. Given that our study aimed to compare the emerging themes between the SAD and the control groups, common themes were identified and compared using statistical methods. The percentages of the responses were compared between groups using the Chi-square test and logistic regression analysis, as per the convergent mixed methods approach, which allows in-depth interview data to be analyzed qualitatively and quantitatively, using statistical methods [24,25,26]. Only the top themes belonging to each time trajectory were included and discussed in this article. In addition to grouping responses related to the expectancies of the effects of cannabis use (like presented in MEEQ domains), other themes unrelated to the direct effects, but related to cannabis use perceptions, were also noted. For instance, the theme ‘morally wrong’, and ‘moderate use’ indicate cannabis-related perceptions, rather than the expectancies of the effects of cannabis. The saturation point related to the themes discussed during the interviews was reached with 12 participants per group, as no new additional information was shared while interviewing our last group of participants. Henceforth, this sample size was sufficient for our study.

## 3. Results

### 3.1. Demographic and Clinical Characteristics

The demographic and clinical characteristics of 26 individuals with SAD and 26 control participants are listed in Table 1. As shown in Table 1, no statistically significant differences were observed between the two study groups, except for the age of the onset of cannabis use, which was earlier (16.6 ± 3.2 mean years of age) for the SAD group compared to the controls (20.1 ± 4.7; *p* ≤ 0.05). Moreover, the lifetime prevalence of CUD in the SAD group was significantly higher. Similarly, SAD participants consumed a greater amount of weekly cannabis and exhibited more problematic cannabis use as per CUPIT scores. Although participants with co-morbid substance and alcohol use disorders were excluded from the study, no significant differences between groups were observed when comparing the alcohol use frequency.

Moreover, our analysis of the use of other recreational substances showed that 2/26 (7.7%) of participants with SAD and 2/26 (7.7%) of the control group used sedatives (χ^2^ = 0; *p* = 1.0). Of those four, three reported using them no more than 10× per month, while only one participant in the SAD reported using benzodiazepines in the last month, but at a frequency of no more than 10 times. The stimulant use was observed in 3/26 (11.5%) in participants with SAD and 4/26 (15.4%) (χ^2^ = 0.17; *p* = 0.685) in the control group, which was mostly consumed no more than 10×/month in the past. Of those, one participant in the control group admitted to using crystal methamphetamine in the past month; however, he used it no more than 10×. The lifetime cocaine use was reported in 8/26 (30.8%) participants in the SAD group and 6/26 (23.1%) in the control group (χ^2^ = 0.39 *p* = 0.532). Of the 14, 13 participants did not use it currently (in the past month) or more than 10×/month at any time in the past. One participant from the SAD group, however, admitted to using cocaine in the past month, but no more than 10×. Only one participant from each group (3.8% in SAD and 3.8% in the control group; χ^2^ = 0; *p* = 1.0) indicated using opioids, but neither one of them was using currently, nor 10×/month in the past. For dissociative anesthetics, there were no current reports of the use, although 2/26 (7.7%) in the SAD group and 1/26 (3.8%) in the control group reported (χ^2^ = 0.35 *p* = 0.552) the lifetime use of no more than 10×/month. Hallucinogens, specifically mushrooms, were the most popular illicit substance among both groups. In the SAD group, 12/26 (46.2%) and 10/26 (38.5%) of controls (χ^2^ = 0.32 *p* = 0.575) reported lifetime use, of which 3/26 (11.5%) of SAD and 2/26 (7.7%) used hallucinogens in the past month, but no more than 10×. Only three participants in the SAD group and four in the control group were regular cigarette users who met the criteria for nicotine use disorder. Considering that the use of substances listed above was overall infrequent and the current use was minimal, their impact on the expectancies of the effects of cannabis use was not assessed, except for alcohol.

### 3.2. Expectancies of the Effects of Repeated Cannabis Use

A cross-sectional comparison of MEEQ domains between SAD and control participants is displayed in Figure 1. This between-group comparison is equivalent to the ‘after repeated use’ stage described in the qualitative analysis.

No statistically significant between-group differences were observed in all three positive domains of MEEQ, including Relaxation and Tension Reduction (F = 0.001; *p* = 0.974; η = 0.0), Social and Sexual Facilitation (F = 1.60; *p* = 0.212; η = 0.031), and Perceptual and Cognitive Enhancement (F = 0.31; *p* = 0.720; η = 0.003); two negative domains, including Cognitive Behavioral Impairment (F = 0.46; *p* = 0.503; η = 0.009) and Global Negative Effects (F = 0.11; *p* = 0.744; η = 0.002); a neutral domain, which was Craving, and Physical Effects (F = 1.10; *p* = 0.300; η = 0.021); and two higher-order domains, higher order Negative (F = 0.15; *p* = 0.700; η = 0.003) and Positive effects (F = 0.04; *p* = 0.838; η = 0.001). Both groups of participants generally responded positively to the effects of relaxation and tension reduction, craving and physical effects, and negatively to the global negative effects of cannabis.

When entering SAD status, sex, age, and current CUD as predictors of individual MEEQ domains (dependent variables) in the multiple regression analysis, only a younger age significantly contributed to the variance of Cognitive and Behavioral Impairment (β = −0.45; *p* ≤ 0.05), Craving and Physical Effects (β = −0.44; *p* ≤ 0.01), and higher order Negative Effects (β = −0.44; *p* ≤ 0.01). Table 2 displays the results of the regression analysis. Next, we conducted the multiple-regression analysis to demonstrate which variables predicted e problematic cannabis use in Table 3. Having SAD predicted problematic cannabis use on each separate occasion when entered together with sex, age, and a selected MEEQ domain. None of the MEEQ domains predicted problematic cannabis use, except for Relaxation and Tension Reduction (β = 0.37; *p* ≤ 0.01) together with SAD (β = 0.43; *p* ≤ 0.01).

Similarly, the bivariate correlation between demographic and clinical variables within the SAD group (shown in Table 4) indicated that age was negatively correlated to Cognitive and Behavioral Impairment, Craving and Physical Effects, higher-order Negative Effects, and total MEEQ scores. The table also demonstrated a positive correlation between current CUD and weekly cannabis consumption, current CUD and the weekly amount of cannabis consumption, and current CUD and CUPIT. Being single was correlated to the MEEQ domains of Relaxation and Tension Reduction, Social and Sexual Facilitation, Perception and Cognitive Enhancement, higher-order Positive Effects, and total MEEQ scores. Education was positively correlated to Craving and Physical Effects while being white was associated with higher weekly cannabis consumption. Interestingly, having lower income within the SAD group was associated with higher problematic cannabis use, as per CUPIT.

Table 5 shows the bivariate correlation between clinical variables for participants with SAD. The total LSAS scores were positively correlated to LSAS Anxiety and Avoidance subscales and the Relaxation and Tension Reduction MEEQ domain. In addition to being correlated to the LSAS Avoidance subscale, the LSAS Anxiety subscale was also positively correlated to the Relaxation and Tension Reduction, Social and Sexual Facilitation, and Higher-order Positive Effects MEEQ domains. LSAS Avoidance was associated with CUPIT, whereas CUPIT was also positively correlated to weekly cannabis use frequency and the amount of cannabis consumption. Weekly cannabis use frequency and the amount of cannabis used were negatively correlated to the Craving and Physical Effects MEEQ domain. A positive correlation was also found between three positive and two negative MEEQ domains, and between the total MEEQ scores and all other individual MEEQ domains.

### 3.3. The Trajectory of Cannabis Use Perceptions Based on Qualitative Analysis

When comparing perceptions about cannabis use between individuals with SAD and the control participants, the findings were categorized into the three following stages: perceptions about cannabis before starting, immediately after consuming cannabis for the first time, and 8–11 years on average after repeated use (Figure 2). Themes corresponding to each stage of cannabis use are described in greater detail below.


*Stage I: Cannabis use perceptions before the initial use*


The emerging themes describing the perception of cannabis before the initial use were both classified as positive and negative. The negative perceptions were that cannabis was impairing, immoral, and addictive, whereas the positive views were that cannabis was enhancing, and used for coping, relaxation, and social facilitation. The percentages of these themes occurring in each group are displayed in Figure 2a. Compared to the controls, the odds of each occurring perception in the SAD group were 0.46 (95% OR CI 0.04–5.81; χ^2^ = 0.39, *p* = 0.0.534) for impairment, 0.47 (95% OR CI 0.08–2.66; χ^2^ = 0.76, *p* = 0.385) for cannabis use being viewed as immoral, 0.51 (95% OR CI 0.10–2.59; χ^2^ = 0.67, *p* = 0.413) addictive, 1.5 (95% OR CI 0.25–8.84; χ^2^ = 0.20, *p* = 0.653) enhancing, 3.57 (95% OR CI 0.53–23.95; χ^2^ = 1.86, *p* = 0.173) used for coping and relaxation, and 3.57 (95% OR CI 0.53–23.95; χ^2^ = 1.86, *p* = 0.173) social facilitation.

The impairment theme at this stage included cognitive, behavioral, and physical types of impairments. Participants with SAD felt that cannabis use would cause damage to their lungs, stunt growth, make them feel tired and hungry, or make them ‘have munchies’. They also reported believing that cannabis would cause negative effects on memory, forgetfulness, dizziness, and disabling inebriation. A 23-year-old female with SAD said the following:


*“I thought it might make me a lot more inebriated and unable to hold a conversation or something or I also maybe just thought it would be more of an intense high where you’re very out of reality.”*


They expected that, over time, they would become ‘lazy’, ‘stupid’, ‘dumb’, ‘degenerate’, ‘goofy’, and ‘out of control’. For instance, according to a 42-year-old male participant with SAD,


*“The only thing I knew about it before using it [cannabis] was stuff I’d see in movies like Cheech and Chong, or stuff like that, or just sitting around being stupid.”*


Before its use, cannabis was believed to be a hard drug, as addictive as cocaine, with the potential to ‘kill you’. Before starting to use cannabis, and like their respective controls, individuals with SAD believed that only ‘bad, cool, rebel kids, and punks’ used cannabis.

‘Hippies’, ‘criminals’, ‘hooligans’, and ‘gangsters messing their lives’ also used cannabis, as cannabis consumption was ‘making them do bad things’. Interestingly, the negative views related to cannabis were influenced by participants’ parents, schoolteachers, and the media.

Despite negative expectancies and perceptions of cannabis use, participants in both groups also understood from their peers and older siblings (already engaging with cannabis) that it would make them more ‘fun’, ‘creative’, ‘happy’, ‘giggly’, and ‘more spiritual’, and that they would ‘laugh a lot’ and generally enjoy using it, due to its enhancing properties. A 32-year-old male described his perceptions as the following:


*“I just thought I was going to laugh. I just thought I was going to laugh and have a good time…”*


Moreover, they felt that cannabis was the social facilitation tool, as it was viewed as ‘something that people do at parties’, as it was ‘lots of fun to do with other people’. Before initiating use, they believed that cannabis was to be used to celebrate a moment with friends and that using it would make them more talkative in socially engaging. They also believed it was something that college kids did in social situations. Lastly, unlike the control group, individuals with SAD believed that cannabis could potentially help with their SAD symptoms and that those affected with mental health conditions may be using it as a ‘coping mechanism’. However, like their respective controls, cannabis was also expected to have relaxing properties.


*Stage II: Cannabis use perceptions immediately after the initial use*


Figure 2b shows the percentages of perceptions about cannabis use at this stage. Aside from the negative views based on which cannabis was recognized as impairing, it was mostly favorably viewed to cause enhancement, coping, relaxation, to be harmlessness, and used for social facilitation. Compared to the controls, the odds of each occurring perceptions in the SAD group were 1.0 (95% OR CI 0.20–4.96; χ^2^ = 0.00, *p* = 1.00) for impairment, 1.40 (95% OR CI 0.28–7.02; χ^2^ = 0.17, *p* = 0.682) for enhancement, 5.00 (95% OR CI 0.75–33.21; χ^2^ = 0.18, *p* = 0.078) for coping and relaxation, 1.43 (95% OR CI 0.27–7.52; χ^2^ = 0.18, *p* = 0.0.673) for perceptions of harmlessness, and 1.00 (95% OR CI 0.16–6.35; χ^2^ = 0.00, *p* = 1.00) for social facilitation.

Individuals with SAD expressed that the enhancing properties of cannabis were as they expected. They reported feeling ‘uplifted’, ‘having fun’, ‘having more intelligent thoughts’, ‘being giggly’, ‘feeling excitement, euphoria’, and being able to enjoy music more and to meditate. Things seemed funnier too. A 22-year-old male reported that the enhancing properties of cannabis exceeded his expectations:


*“So I guess I just was enlightened to know what it actually meant, and what the euphoria was. So my perception became. This isn’t a bad thing. This is a thing that can make me feel happy, and giggly, and what not.”*


Moreover, SAD participants’ expectations were met, as after using cannabis, they were able to cope with social situations, relax, and to enjoy social settings. They felt more peaceful and calmer when together with friends. Smoking cannabis was what ‘cool stoner kids did’. One participant reported benefits for his SAD symptoms, which, again, exceeded expectations:


*“I’m not staying up anymore thinking about whatever happened earlier in the day and I’m distracted. I can focus on my Netflix or whatever, or I can take the notes from this lecture much easier. If that makes sense.”*


Contrary to these findings, the participants in the control group focused more on the relaxing properties of cannabis, which helped them unwind both at home and in social situations. The emerging theme at this stage is the perception of harmlessness related to the effects of cannabis. After trying it for the first time, individuals with SAD reported that cannabis was ‘not so bad, or ‘not too intense’ on health and functioning than they had originally imagined. Smoking cannabis was ‘not a big deal’ as originally believed. A 22-year-old male expressed previously being misinformed regarding the effects of cannabis when he stated the following:


*“But I learned that, that was quite actually misinformation. It’s not the whole truth of it, eventually. I thought that it was something that not serious people did, I guess…. So I was like. It’s not a big deal. All these people are doing it, and they show up at school the next day. What’s the big deal? So I think it just became disillusioned to me, as not a big deal, it wasn’t a hard drug. It’s not like they were doing cocaine or anything. It was just a joint.”*


Aside from the mainly positive expectancies of the effects of cannabis, participants with SAD also reported cognitive, behavioral, and physical impairments. Unlike in the first stage, in which their perceptions were based on what was reported by others, these perceptions were based on their own experiences. Several participants indicated that cannabis extenuated their state of mind, so instead of making them feel better, cannabis use made them feel worse when they were down. Accordingly, a 28-year-old female described her experience as follows:


*“So the first time that I did it, we were planning to go out and I thought to myself, okay, it makes you free, or it makes you less stuck in your head. If I do it, maybe I’ll be able to go out with my group, maybe have a fun time, but that didn’t happen. It just elevated my anxiety even more, and I had to go back home.”*


Like the 28-year-old participant describing her experiences above, some participants unexpectedly felt anxiogenic instead of anxiolytic effects, such as elevated anxiety, palpitations, sweating, feelings of paranoia, and panic.


*Stage III: Cannabis use perceptions after repeated use*


Figure 2c displays cannabis use perceptions after repeated use. At this stage, the views related to cannabis were both negative and positive. Participants from both groups expressed concerns about the addictive and impairing properties of cannabis, as well as cautioning about the need to moderate cannabis use. However, simultaneously, cannabis was also favorably viewed to boost coping, relaxation, and enhancement. Compared to the controls, the odds of each occurring motivation in the SAD group were 3.0 (95% OR CI 0.53–16.9; χ^2^ = 1.62, *p* = 0.203) for addiction, 1.43 (95% OR CI 0.27–7.52; χ^2^ = 0.18, *p* = 0.673) impairment, 0.72 (95% OR CI 0.14–3.58; χ^2^ = 0.17, *p* = 0.682) for moderating use, 3.00 (95% OR CI 0.53–16.9; χ^2^ = 1.62, *p* = 0.682) for coping and relaxation, and 0.24 (95% OR CI 0.04–1.36; χ^2^ = 1.81, *p* = 0.094) for enhancement. Like in the second stage of cannabis use, participants with SAD expressed that cannabis was helping them relax and deal with social situations. They felt that cannabis was helping them with social anxiety and anxiety in general. Contrary to these findings, participants in the control group mainly reported relaxing and unwinding with cannabis. At this stage, participants with SAD also reported enhancing properties, as cannabis was helping them ‘self-reflect’ and ‘enjoy’ being in various situations.

The theme of physical, cognitive, and behavioral impairment reappears at this stage, as well with reports of the anxiogenic effects of cannabis. In addition to what was already reported prior and immediately after using cannabis, several participants with SAD reported that cannabis was making them feel ‘less motivated’ and ‘unambitious’. In fact, a 32-year-old male participants with SAD quoted the following:


*“I would have graduated school 10 years ago. It [cannabis use] is kind of dangerous like that.”*


As in the first stage, the theme of addiction reoccurs after repeated cannabis use; however, this time, participants in both groups focused on their personal experiences with cannabis craving, tolerance, and withdrawal, or what they have observed when using cannabis with their peers. For instance, a 23-year-old female, who uses cannabis to manage her social anxiety, stated the following:


*“I think I have to smoke more now to manage the anxiety than I did before. It’s really something that it’s more of a cyclical effect”*


Similarly, a 27-year-old male with SAD stated


*“And yeah, I’m more conscious of cannabis’ addictive potential, because before I thought it wasn’t addictive, so no. I’m more aware that it is addictive. Yeah, like the effect is not the exact same as the first day….Then, I used to use it because it takes a little bit more, but before I used to use 0.75 grams in one day. Now I use 1 gram.”*


Lastly, at this stage, a new theme emerged, which was titled ‘moderate use’, according to which both groups of participants described having to adjust and reduce their cannabis use over time. Participants with SAD described using ‘too much’ cannabis ‘too often’, and being ‘overwhelmed’ with the effects of cannabis at the early stages of consumption, which was causing function impairment. For instance, a 24-year-old female with SAD reported the following:


*“I also think it can be abused and I think it can be harmful in a way to people….You should try to reduce, like once in a while is good”*


Some participants felt that cannabis is not for everyone, while one participant expressed an opinion that cannabis should not be used until an individual is 18 years of age, because, by watching his peers, he felt that responsible cannabis use requires maturing and knowledge about how to use cannabis properly. A few subjects with SAD also discussed moderating use, in terms of finding and using the right strain of cannabis. For instance, according to a 28-year-old female with SAD,


*“You have to find the right fit for yourself, like and what strains to do, because not all the effects… Something can be enjoyable for one person and have a really stressful effect on somebody else. …So right now I feel like you should make an informed decision about whatever you’re going to try and don’t force it on yourself, something just because everyone else was doing it…..I know what strains I like now, and what strains might give me more anxiety than I already have.”*


Like the 28-year-old participant describing the adjustments she made with her cannabis use, some participants recognized that some strains may elicit calmness, whereas others have anxiogenic effects on their social anxiety.

## 4. Discussion

### 4.1. Expectancies of the Effects after Repeated Use

The primary objective of our study was to compare the expectancies of the effects of cannabis use between individuals with and without SAD. Our data showed no significant differences across all six lower-order and two higher-order MEEQ questionnaire domains between the two groups. Contrary to the work conducted by Buckner and colleagues [9,10], the MEEQ assessment indicated that individuals with SAD did not have higher expectancies of global negative effects or cognitive-behavioral impairments with cannabis use. Moreover, the average scores for both groups demonstrated higher expectations of relaxation and tension reduction and craving and physical effects, as well as higher-order positive effects. Our findings were consistent with the expectancy theory [18] and previous studies [19,20], indicating that, regardless of their psychiatric history, frequent cannabis users, such as those recruited in our study, were more likely to report positive and less likely negative expectancies of cannabis use. Accordingly, these positive expectancies may have driven repeated cannabis consumption, regardless of the experienced consequences, such as developing cannabis use addiction or dependence.

Our regression analysis revealed that age was negatively correlated to cognitive and behavioral impairment and global negative effects. Moreover, we showed that, within the SAD group, age was negatively correlated to cognitive negative impairment and higher-order negative effects. Our finding may explain the results produced by the previous studies, showing occurrent negative expectancies in emerging adults with SAD with narrow age groups [9,10]. High school students and young college students with SAD are often exposed to unique social settings of their vocational environments, in which they are constantly surrounded by groups of people. Such settings may be social anxiety-triggering [2,3,4,5], prompting these individuals to feel like they are benefiting from cannabis-related impairments [9,10]. They may use cannabis as a ‘camouflage’ for their self-perceived inadequate social behavior, which they may attribute to cannabis impairment, rather than their personality [9,10,18]. However, adults with SAD (like participants from our study) are no longer exposed to such social situations, as they are no longer in school, and may no longer need to use cannabis to cover up their self-perceived behavior.

Craving for cannabis was more likely to be reported by SAD participants who consumed less cannabis and at a lower frequency of use. These findings contrast the current body of evidence, indicating that more cannabis cravings predicted increased use [32,33,34,35]. However, a closer examination of our data from in-depth interviews indicated that, after repeated cannabis use, participants attempted to mindfully reduce their cannabis consumption. The attempts were made to mindfully curb cannabis consumption by resisting the impulse to use, which would have otherwise satisfied their cravings.

The association between expectancies of cravings and reduced cannabis use may be explained by recent research [34,36,37] on the role of psychosocial strategies which influence the relationship between craving and cannabis use. Some of these described strategies included mindfulness about cannabis use. The research suggested that mindfulness moderated the relationship between craving and the frequency of use [34]. In one study, it was shown that being mindful of craving for substances [36] reduced the likelihood of relapse following treatment for substance use [38]. Accordingly, our participants reported being cognizant of their addiction and craving impulses, as they knew not to act upon them to consciously reduce their substance use [37,39].

After controlling for covariates, our data showed that tension reduction expectancies and SAD separately predicted problematic cannabis use. Moreover, in the SAD group, tension reduction was more pronounced in those with higher severity of illness. Expectancies of tension reduction, social and sexual facilitation, and higher-order positive effects were more expressed by those with higher SAD severity and participants living alone without a partner. Considering that previous research indicated that living in isolation and without family support may lead to functional impairment [2,3,4,5], and that scoring high on the positive expectancy domains of MEEQ was correlated to poorer psychosocial functioning [40,41], our finding that single rather than married participants with SAD scored higher on positive expectancies were not unusual, possibly because single individuals were more likely to use cannabis than the support of others to deal with life situations.

Although no differences were observed between regular cannabis users with and without SAD when assessing MEEQ Tension Reduction and Relaxation domain scores, the findings from in-depth interviews showed the differences in the types of tension the two groups were expecting to relieve. While the non-SAD group expected unwinding from the stress caused by the situation of daily living, those with SAD expressed expecting to release tension and cope with social anxiety through cannabis use. The notion of using cannabis to cope with SAD is supported by the biopsychosocial model of Social Anxiety Disorder and Substance Use Disorders [42,43], which states that socially anxious individuals may use cannabis to cope with SAD, and they may persist in doing so, despite developing CUD. In line with this model, our data demonstrated the conscious decision to repeatedly use cannabis in the SAD group, despite the high current CUD rates of 57.7%.

### 4.2. The Time Trajectory of Cannabis Use Perceptions

Our secondary objective was to examine cannabis-related perceptions before the initial use, immediately after the initial use, and several years after repeated cannabis use between individuals with and without SAD, using in-depth interview open-ended questions. Although no between-group differences were found in the emerging themes, it is noteworthy that participants with SAD started using cannabis significantly earlier (age: 16.6 ± 3.2) than participants without (age: 20.1 ± 4.7). The age of the onset of cannabis use for participants without SAD is consistent with that reported by Health Canada in the Canadian Cannabis Survey, 2022 (20.5 years of age), which assessed cannabis use trends from 10,048 Canadians [44], therefore, our non-SAD sample served as a valid comparison to people with SAD.

Before initial cannabis use, both negative and positive perceptions emerged during the interviews. Cannabis was viewed as a dangerous drug, highly addictive, and bound to cause several physical, cognitive, and behavioral problems. Given that most started their consumption before the Canadian legalization of cannabis for recreational purposes, the emerging negative views were unsurprising and consistent with the expectancy theory [18], as well as with previous research [9,20] indicating that non-users rather than regular users tended to perceive cannabis more negatively. Contrary to the negative, positive cannabis views were also reported by both groups before initiating the consumption. The top three positive views included enhancement, coping and relaxation, and social facilitation. Participants with SAD also reported expecting that cannabis assisted them as a coping mechanism for their SAD, possibly prompting much earlier consumption than their respective controls combined with their higher motivations for coping and enhancement [14].

In the second stage, immediately after trying cannabis for the first time, a significant shift towards positive expectancies occurred in both groups. Aside from the expected cognitive, behavioral, and physical impairments, the emerging themes are the expectancies of enhancement, social facilitation, relaxation, and the perception that cannabis is harmless. Seemingly, as opposed to being viewed as a dangerous drug, at this stage, the effects of cannabis were generally positive after a single use, possibly enough to drive repeated use.

The positive cannabis-related views could be explained by the pharmacological effects of two major components in cannabis: δ-9-tetrahydrocannabinol (THC), and cannabidiol (CBD). CBD is a non-psychomimetic substance demonstrated to have anxiolytic properties. In cross-sectional and longitudinal randomized, placebo-controlled studies, 300 mg–600 mg of CBD was shown to be effective in reducing symptoms of SAD [45,46,47,48,49]. Conversely, in human studies, THC was found to have anxiogenic effects [50,51,52]. Furthermore, THC was noted to induce dopamine release, promoting feelings of euphoria and excitement [53,54].

Consequently, after trying cannabis for the first time, participants in our study reported feeling relaxed, calm, and less distracted by their symptoms of SAD. They reported being able to enjoy social situations more because of the perceived calming effects of cannabis they were experiencing. These effects may have been experienced due to consuming CBD in their cannabis. However, some participants also expressed unpleasant symptoms of anxiety, which may have been experienced due to the anxiogenic properties of THC. THC-stimulating properties may have resulted in the enhancing experiences both groups described after initial use.

At the final stage of cannabis use, the perceptions shifted again towards positive and negative expectancies emerging concurrently. In terms of the positive perceptions, the reported themes were enhancements, coping, and relaxation. Conversely, the symptoms of tolerance, withdrawal, and impairments in daily functioning were also reported. The negative perceptions could be explained by the effects of THC in cannabis that occur due to chronic exposure [53,54]. After chronic exposure, the reduced ability of stimuli to activate reward neurocircuitry is noted due to decreases in dopamine receptor density [53,54]. Given that, at this stage, reward circuitry can no longer be activated by consuming the same amounts of THC in cannabis, chronic users may develop tolerance, prompting them to start consuming higher amounts of cannabis to experience the same desired effect/euphoria as before [55,56]. In the absence of drug use, chronic users start experiencing symptoms of pharmacological withdrawal due to the prolonged suppression of the reward circuits in the brain, leading to anhedonia, a general state of depression, and a craving. To evade the negative feelings, chronic users begin using cannabis in higher amounts and at a greater frequency [53,54,55,56]. Henceforth, at the last stage of cannabis use, our participants continued using cannabis, not only due to reported positive experiences, but possibly because they needed to avoid feeling the debilitating symptoms of withdrawal.

A closer examination of the participants’ demographic and clinical characteristics related to the cannabis use perception trajectory, data could provide some insight into the patterns of problematic cannabis use. Although both groups of participants reached their peak of cannabis use consumption at 23 years of age (23.1 ± 8.5 for the SAD group; 23.0 ± 4.5 for the controls), the exposure to cannabis is much longer for those with SAD, given the earlier age of onset. The longer exposure to cannabis in SAD, including initiating cannabis use earlier in teens rather than in one’s early twenties, was previously linked to a greater risk of developing problematic use [57,58]. Therefore, at the peak of cannabis use consumption, it was not surprising to observe significantly higher incidences of lifetime CUD in SAD (80.8% for SAD and 53.8% for the controls; *p* < 0.01).

Approximately 3–4 years after reaching the peak for cannabis use consumption, the incidences of CUD significantly dropped (57.7% for SAD and 46.3% for the controls), and even more so in SAD. The observed drop could be explained by participants’ need to moderate their use, reported at the last stage of the cannabis perceptions trajectory. Accordingly, some participants expressed having to reduce their frequency and amount of consumed cannabis, due to the functional disability they experienced related to cannabis use. Nevertheless, the incidences of CUD at the time of data collection remained alarmingly high for both groups, at which point, participants chose to maintain their cannabis consumption, despite having numerous negative cannabis-related experiences.

### 4.3. Scientific Significance

Cannabis use reduction and prevention efforts could be informed by carefully examining the trajectory of cannabis use perceptions demonstrated in our research. Implementing psychoeducational programs that would inform adolescents about the likelihood of this ‘downward spiral’ might be beneficial in preventing the onset of cannabis use. For example, one such program tested in controlled randomized trials is the Preventure program, designed to target adolescents at risk of engaging in alcohol and substance use [59,60,61]. The risk assessment is based on identifying personality traits (e.g., hopelessness, anxiety sensitivity, impulsivity, and sensation seeing). Once identified, the high-risk individuals receive interventions held during school hours, which include personality-tailored psycho-educational, motivational enhancement, and cognitive behavioral therapy. The data from clinical trials indicate that the Preventure program was effective in reducing alcohol and substance use harm by 50% [59,60,61]. Similar programs designed to identify SAD in schoolchildren younger than 16 years of age when they are at risk of starting cannabis use could be useful. Bringing awareness to the negative impacts of SAD and its consequences later in life to adolescents, their parents, and teachers in a non-stigmatizing manner may aid in early SAD diagnosis and treatment, instead of using cannabis to cope with SAD.

Specific prevention and harm reduction efforts could also be implemented to benefit adults with SAD. For instance, primary healthcare providers should engage in discussing cannabis use with non-using clients with SAD or suspected SAD. The discussion could acknowledge the perceived anecdotal benefits of cannabis on symptoms of SAD while balancing the explanation of perceived immediate versus long-term benefits and harms (such as cannabis addiction) [62]. Effects of cannabis and its components on polypharmacy and drug–drug interactions could be assessed, specifically in older adults and those affected by multiple physical and/or other psychiatric conditions. Cognitively impaired individuals should be aware of the additional adverse effects they may experience due to cannabis consumption, should they choose to do so [62,63]. The harm reduction strategies for those with SAD could focus on reducing the negative consequences associated with cannabis use. For instance, discouraging cannabis use in situations where its consumption may pose a risk (e.g., while driving a car, caring for a child, before work, or in public places), or using small amounts of cannabis gradually and seeing its effects to prevent significant intoxication could also be considered [62]. Prompting individuals with SAD to pay more attention to the types of ratios of cannabinoids on product labels could also be beneficial, especially given that it has previously been shown that higher THC in cannabis was linked to anxiogenic effects in those already suffering from anxiety-related symptoms [62,64]. Most importantly, clinicians and mental health workers could specifically be made aware of the symptoms of SAD in adolescents and adults to encourage rapid diagnosis and the administration of conventional treatments for SAD.

### 4.4. Limitations

The present study has several limitations, which may be addressed in future work in this area. First, our qualitative data on the trajectories of cannabis use were reliant on the recalls and memories of these events, as they happened in the past. Given that this data collection strategy is susceptible to memory bias, longitudinal studies investigating how cannabis use perceptions change over time could be conducted to minimize such bias. Our cohort mainly consisted of participants who were not conventionally treated, as we felt that our sample should be aligned with most of the SAD population. Consequently, the experiences may be different from the clinical sample of those with SAD. Furthermore, our study focused on examining cannabis use perceptions in regular cannabis users with and without SAD. Future studies could expand to study perceptions of regular users, as well as infrequent cannabis users and non-users to determine the protective factors, opinions, and individual characteristics associated with abstinence. Our study focused on examining the expectancies of the effects of cannabis in individuals both with and without SAD, who did not meet the criteria for alcohol use disorder, or any other substance use disorders. Although we demonstrated no correlation between alcohol-use frequency and demographics, clinical characteristics of our participants, and MEEQ outcomes, future studies may assess the contribution of other substance use disorders or alcohol dependence on the expectancies of the effects of cannabis use in SAD. The consideration of performing such an assessment is due to the reported polysubstance use in individuals with SAD [65,66]. Lastly, we recruited 26 participants per group, which allowed for the adequate comparison of the expectancies of the effects of cannabis use at a cannabis use maintenance stage. Although our sample size was sufficient to assess the impact of major covariates such as age, sex, and current CUD, a larger sample size would have allowed us to examine the broader effects of demographic and clinical variables on the expectancy data. Nevertheless, our study provides an important insight into expectancies of the effects of cannabis use in individuals with SAD, which could valuably inform clinicians when planning cannabis-use prevention and risk-reduction efforts.

## 5. Conclusions

By demonstrating no differences related to the expectancies of the effects of cannabis use between adults with and without SAD who used cannabis frequently, and by showing that both groups of participants acknowledged having positive expectancies, our findings were aligned with the expectancy theory. Accordingly, our research indicated that, regardless of their psychiatric history, frequent cannabis users are more likely to report positive expectancies associated with their higher cannabis use patterns. Moreover, we have shown that age and social contexts, rather than the SAD diagnosis alone, may have influenced the reports of negative expectancies in previous research on emerging adults. Seemingly, young adults were more likely to be exposed to unique ‘anxiety-triggering’ social contexts; henceforth, the negative expectancies may have been a byproduct of self-handicapping. In contrast, most adults with SAD, (like most participants in our study) were no longer in school, therefore, attributing their self-perceived inadequacies of their social behaviors to cannabis was no longer needed to protect their self-image.

The acknowledgement of expecting tension reduction and relaxation specifically among SAD participants aligned with our previous study on the same cohort, demonstrating increased cannabis use motivations for coping. Our findings in this study are also in tandem with the Biopsychosocial Model of Social Anxiety Disorder for Substance Use Disorder, which showed that cannabis is used for coping with symptoms of SAD, despite developing problematic cannabis use and CUD. Consequently, the incidence of current CUD among participants with SAD was relatively high, at 57.7% in this study. Furthermore, higher tension reduction expectancies in those with higher SAD severity and those living socially isolated were unsurprising, considering that cannabis was generally described as a self-medicating tool. Those experiencing a greater number of symptoms and those lacking the social support might have been more hopeful to receive the relief they needed from cannabis. Within the SAD group, the negative association between craving and the amount and frequency of cannabis consumption may have been the result of the reported attempts to decrease and moderate cannabis use. The mindful attempts of cannabis use reduction may have led to the significant reduction in observed incidences of CUD from the time of the highest cannabis use consumption to when the interviews were administered.

The data on the trajectory of cannabis use perceptions indicated that individuals with SAD started using cannabis significantly earlier than their respective controls, putting them at risk of developing higher incidences of CUD. Before using cannabis, both groups had both negative and positive perceptions of cannabis use; however, it appears that the positive views may have prompted the onset of cannabis use. Immediately after the initial use, a general shift towards positive cannabis perceptions was seen. Most significantly, the cannabis view shifted from being harmful to completely harmless to human well-being, and the predominantly positive perceptions may have resulted in continuous cannabis consumption. Despite experiencing signs of declining function, and possibly due to symptoms of addiction, our study participants with and without SAD insisted on maintaining their cannabis consumption.

Our research supports the idea that psychoeducational and cannabis use prevention efforts should target schoolchildren in their early adolescent years, informing them of the expected cannabis use perceptions trajectory and the risks associated with repeated cannabis use. Moreover, open discussions about the risks of cannabis use with adults with SAD planning to use cannabis should be encouraged. Also, having an open conversation about harm-reduction strategies could be promoted with those already using cannabis, while aiming to reduce the risk of cannabis consumption. Informational programs aiding the identification, diagnosis, and treatment of SAD should also be implemented in vocational settings and locations where primary healthcare is offered. Such campaigns could prevent resorting to cannabis use instead of seeking more conventional treatments for SAD.

## Figures and Tables

**Figure 1 brainsci-14-00246-f001:**
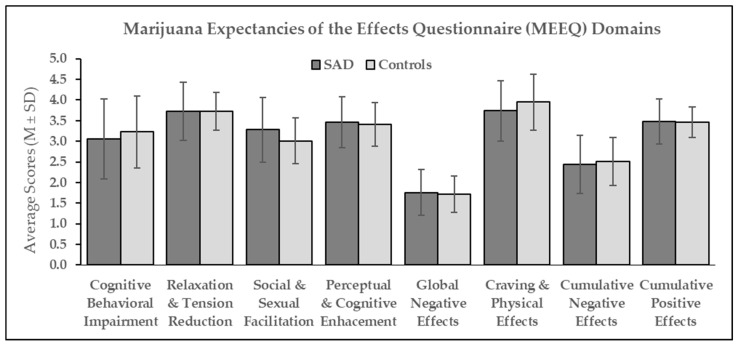
Cross-sectional comparison of Marijuana Expectancies of the Effects (MEEQ) domains. SAD = Social Anxiety Disorder; M = mean; SD = standard deviation; (n = 26 for the SAD and n = 26 for the control group).

**Figure 2 brainsci-14-00246-f002:**
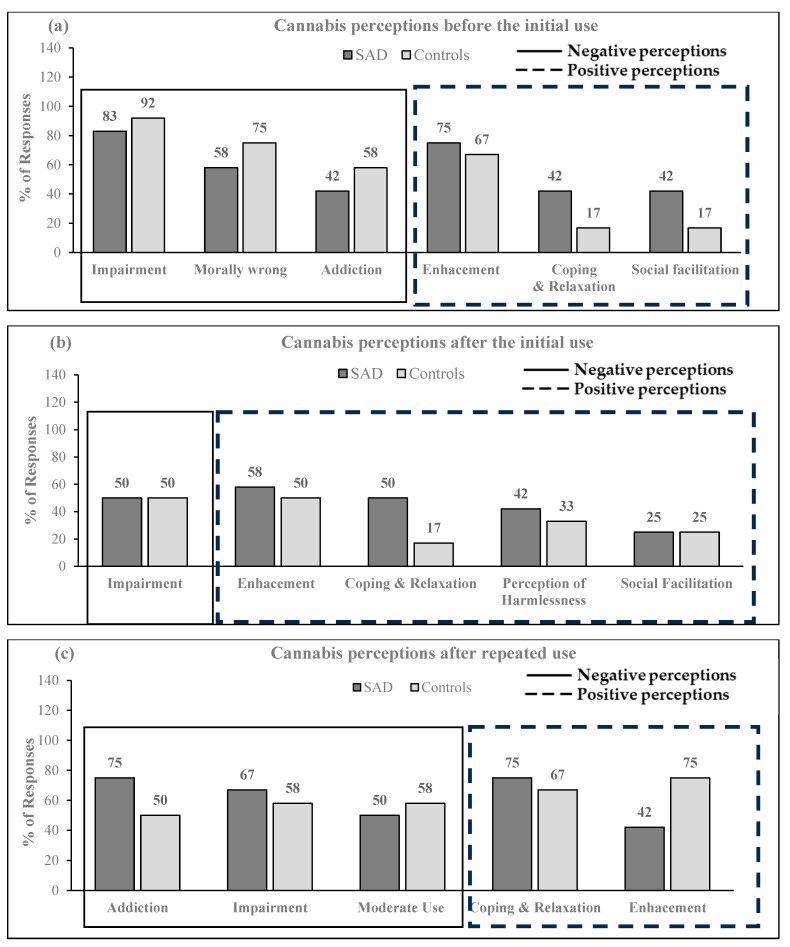
Cross-sectional comparison of the perceptions of cannabis use for n = 12 in SAD and n = 12 in the control group. (**a**) Cannabis perception before the initial use, (**b**) Cannabis perceptions after the initial use, (**c**) Cannabis perceptions after repeated use. SAD = Social Anxiety Disorder; % = percentage. Themes inside solid lines indicate negative perceptions, whereas themes inside dashed lines are positive perceptions.

**Table 1 brainsci-14-00246-t001:** Demographic and clinical characteristics of study participants.

	SAD Group (n = 26)		Control Group (n = 26)			
	M or %	SD	M of %	SD	t or χ^2^	d or Phi
**Age (years)**	27.92	7.34	27.35	6.69	0.30	0.08
**Age of onset of cannabis use**	16.63	3.15	20.12	4.74	−3.12 *	0.87
**Age of highest cannabis use**	23.05	8.48	23.02	4.50	0.02	0.01
**Sex (% female)**	53.8		50.0		0.08	0.04
**Race (% Caucasian)**	76.9		57.7		2.19	0.21
**Education (% university or higher)**	46.2		65.4		1.95	0.19
**Employed or student (%)**	73.1		92.3		3.36	0.25
**Last year’s income ≥ %50,000**	40.0		66.7		3.50	0.27
**Married/common law (%)**	23.1		19.2		0.12	0.05
**Cannabis use (times/week)**	5.06	2.19	3.96	2.28	1.77	0.49
**Cannabis use (g/week)**	7.57	9.37	2.00	1.81	2.92 ^†^	0.82
**Alcohol use (drinks ^1^/week)**	2.22	3.78	1.93	1.85	0.35	0.10
**The severity of SAD (per total LSAS)**	92.65	20.82	9.46	6.52	19.45 ^‡^	5.39
**LSAS Anxiety scores**	48.92	9.63	6.27	4.58	20.39 ^‡^	5.66
**LSAS Avoidance scores**	43.73	12.05	3.19	3.46	16.49 ^‡^	4.57
**Cannabis problem severity ** **(per CUPIT)**	36.54	12.35	26.31	9.80	3.31 ^†^	0.92
**SAD comorbidities (%):**						
**Current CUD**	57.7		46.2		0.70	0.12
**Lifetime CUD**	80.8		53.8		4.28 ^†^	0.29
**Past MDD**	34.6					
**Current GAD**	26.9					
**Other ^2^**	30.8					

* *p* ≤ 0.05; ^†^ *p* ≤ 0.01; ^‡^ *p* ≤ 0.001. SAD = Social Anxiety Disorder; CUD = Cannabis Use Disorder; MDD = Major Depressive Disorder; GAD = Generalized Anxiety Disorder; LSAS = Liebowitz Social Anxiety Scale; CUPIT = Cannabis Use Problems Identification Test; M = mean; SD = standard deviation; g = grams; % = percentage; χ^2^ = chi-square test; t = independent *t*-test; d = Cohen’s d; Phi = Phi coefficient. ^1^ One drink is defined as either a 12-oz (341 mL) bottle of 5% alcohol beer or cider, a 5-oz (142 mL) glass of 12% alcohol wine, or a 1.5-oz (43 mL) shot glass of 40% alcohol spirits. ^2^ Other co-morbidities: Past Alcohol Use Disorder, Other Specified Eating Disorder, Current Nicotine Use Disorder, Past Nicotine Use Disorder, Current Persistent Depressive Disorder, Past Persistent Depressive Disorder Past Bulimia Nervosa, Current Major Depressive Disorder, Post-traumatic Stress Disorder Not Otherwise Specified, and Other Specified Depressive Disorder.

**Table 2 brainsci-14-00246-t002:** Variables predicting marijuana effect expectancies in the after-repeated use stage.

	Dependent Variables															
	Cognitive & Behavioral Impairment		Relaxation & Tension Reduction		Social & Sexual Facilitation		Perceptual & Cognitive Enhancement		Global Negative Effects		Craving & Physical Effects		Cumulative Negative Effects		Cumulative Positive Effects	
Predictor variables (n = 52)	B (SE)	β	B (SE)	β	B (SE)	β	B (SE)	β	B (SE)	β	B (SE)	β	B (SE)	β	B (SE)	β
**^1^** **SAD status**	−0.14 (0.24)	−0.08	−0.03 (0.17)	−0.03	0.21 (0.19)	0.15	0.05 (0.16)	0.04	0.05 (0.14)	0.05	−0.17 (0.18)	−0.12	−0.05 (0.16)	−0.04	0.01 (0.13)	0.01
**^1^** **Sex**	−0.14 (0.24)	−0.08	0.05 (0.17)	0.04	0.13 (0.19)	0.10	−0.13 (0.15)	−0.12	0.02 (0.14)	0.14	−0.17 (0.18)	−0.12	−0.07 (0.16)	−0.05	0.06 (0.13)	0.06
**Age**	−0.06 (0.02)	−0.45 ^‡^	0.01 (0.01)	0.07	−0.01 (0.01)	−0.11	−0.02 (0.01)	−0.25	−0.02 (0.01)	−0.26	−0.04 (0.01)	−0.44 ^†^	−0.04 (0.01)	−0.44 ^†^	−0.01 (0.01)	−0.20
**^1^** **Current CUD**	0.07 (0.24)	0.04	0.28 (0.17)	0.24	0.29 (0.19)	0.22	0.26 (0.16)	0.23	0.06 (0.14)	0.06	−0.01 (0.18)	−0.01	0.07 (0.17)	0.05	0.19 (0.13)	0.21

^†^ *p* ≤ 0.01, ^‡^
*p* ≤ 0.001; n = number of participants; MEEQ = Marijuana Expectancies of the Effects Questionnaire; CUD = Cannabis Use Disorder; SAD = Social Anxiety Disorder; B = unstandardized coefficient beta; SE = Standard Error; β = standardized coefficient beta; Asterix and the cross sign indicate statistically significant values based on the independent *t*-test. ^1^ dummy coding for sex (0 = male, 1 = female); current CUD (0 = no, 1 = yes); SAD status (0 = no; 1 = yes).

**Table 3 brainsci-14-00246-t003:** Variables predicting problematic cannabis use in the after-repeated use stage.

Predictor Variables: (n = 52)	Dependent Variable: Problematic Cannabis Use (CUPIT)					
	B	SE	β	R^2^	Adj R^2^	F
**SAD ^1^**	9.92	2.82	0.41 ^‡^	0.60	0.31	**6.67 ^‡^**
**Age**	0.20	0.20	0.17			
**Sex ^1^**	3.98	2.81	0.11			
**MEEQ Tension Reduction and Relaxation**	7.47	2.39	**0.37 ^†^**			

**SAD ^1^**	9.72	3.14	**0.40 ^†^**	0.48	0.23	**3.54 ^†^**
**Age**	0.23	0.23	0.23			
**Sex ^1^**	4.34	3.11	0.18			
**MEEQ Social and Sexual Facilitation**	0.89	2.35	0.05			

**SAD ^1^**	9.81	3.10	**0.41 ^†^**	0.49	0.24	**3.60 ^†^**
**Age**	0.26	0.23	0.15			
**Sex ^1^**	4.67	3.10	0.19			
**MEEQ Perceptual and Cognitive Enhancement**	1.64	2.85	0.08			

**SAD ^1^**	10.26	3.05	**0.43 ^†^**	0.51	0.26	**4.06 ^†^**
**Age**	0.37	0.25	0.21			
**Sex ^1^**	4.81	3.05	0.20			
**MEEQ Cognitive Behavioral Impairment**	2.49	1.89	0.19			

**SAD ^1^**	9.69	3.04	**0.40 ^†^**	0.51	9.26	**4.10 ^†^**
**Age**	0.31	0.23	0.18			
**Sex ^1^**	4.40	3.04	0.18			
**MEEQ Global Negative Effects**	4.38	3.30	0.18			

**SAD ^1^**	9.58	3.10	**0.40 ^†^**	0.49	0.24	**3.71 ^†^**
**Age**	0.13	0.25	0.08			
**Sex ^1^**	4.13	3.10	0.17			
**MEEQ Craving and Physical Effects**	−0.20	2.49	−0.12			

**SAD ^1^**	9.80	3.06	**0.41 ^†^**	0.50	0.25	**3.92 ^†^**
**Age**	0.28	0.23	0.16			
**Sex ^1^**	4.20	3.06	0.17			
**MEEQ Higher-order Positive Effects**	3.96	3.45	0.15			

**SAD ^1^**	10.11	3.02	**0.42 ^†^**	0.52	0.27	**4.28 ^†^**
**Age**	0.39	0.25	0.23			
**Sex ^1^**	4.73	3.02	0.20			
**MEEQ Higher-order Negative Effects**	4.20	2.70	0.22			

^†^ *p* ≤ 0.01; ^‡^ *p* ≤ 0.001; n = number of participants; MEEQ = Marijuana Expectancies of the Effects Questionnaire; CUPIT = Cannabis Use Problems Identification Test; SAD = Social Anxiety Disorder; B = unstandardized coefficient beta; SE = Standard Error; β = standardized coefficient beta; R^2^ = coefficient of determination; Adj. R^2^ = adjusted coefficient of determination; F = explained variance. ^1^ dummy coding for sex (0 = male, 1 = female); SAD status (0 = no; 1 = yes).

**Table 4 brainsci-14-00246-t004:** Bivariate correlation between demographic and clinical measures for participants with (Social Anxiety Disorder (SAD).

(n = 26)	Age	Sex ^1^	Current CUD ^1^	Lifetime MDD ^1^	Current GAD ^1^	Educ. ^1^	Race (WC) ^1^	Race (EA) ^1^	Income ^1^	Occ. Status ^1^	Marital Status ^1^
**Total LSAS**	−0.16	−0.06	0.28	0.24	−0.16	−0.09	−0.18	−0.03	−0.29	−0.26	−0.18
**LSAS Anxiety**	−0.28	0.03	0.30	0.30	−0.11	−0.01	−0.22	−0.06	−0.23	−0.21	−0.24
**LSAS Avoidance**	−0.06	−0.12	0.24	0.17	−0.18	−0.15	−0.14	0.01	−0.31	−0.29	−0.12
**Weekly cannabis use frequency**	0.14	0.21	**0.60 ^†^**	0.18	0.33	−0.31	**0.55 ^†^**	0.34	−0.28	−0.07	−0.06
**The weekly amount of cannabis use (g)**	0.13	0.23	**0.42 ***	−0.01	0.30	−0.38	0.24	0.18	−0.35	−0.09	0.01
**The weekly alcohol frequency (drinks ^2^/week)**	0.16	−0.08	−0.19	−0.12	−0.23	−0.34	0.08	−0.18	−0.02	0.19	0.07
**CUPIT**	0.13	0.16	**0.64 ^†^**	0.19	0.35	−0.35	0.27	0.20	**−0.55 ^†^**	−0.13	−0.21
**MEEQ Total**	**−0.59 ^†^**	0.07	0.19	0.03	0.12	0.34	−0.06	0.05	0.07	0.05	**−0.51 ^†^**
**MEEQ Cognitive & Behavioral Impairment**	**−0.44 ***	−0.17	0.05	−0.12	0.15	0.33	−0.17	−0.07	0.04	0.03	−0.15
**MEEQ Relaxation & Tension Reduction**	0.01	0.18	0.21	0.04	−0.06	−0.18	−0.03	0.12	−0.21	0.09	**−0.52 ^†^**
**MEEQ Social & Sexual Facilitation**	−0.35	0.36	0.21	0.25	0.09	0.07	0.18	0.11	0.04	0.21	**−0.62 ^†^**
**MEEQ Perceptual & Cognitive Enhancement**	−0.35	−0.06	0.12	−0.07	−0.07	0.20	0.17	0.28	0.12	0.12	**−0.53 ^†^**
**MEEQ Global Negative Effects**	−0.37	−0.20	0.02	0.02	−0.07	0.30	−0.15	−0.24	0.05	−0.11	0.04
**MEEQ Craving & Physical Effects**	**−0.44 ***	0.05	−0.03	−0.12	0.02	**0.50 ***	−0.22	0.03	0.23	−0.22	0.13
**MEEQ Cumulative Negative Effects**	**−0.46 ***	−0.20	0.05	−0.12	0.13	0.36	−0.18	−0.15	0.05	−0.02	−0.10
**MEEQ Cumulative Positive Effects**	−0.32	0.37	0.15	0.20	0.04	0.10	−0.12	0.13	0.08	0.08	**−0.52 ^†^**

* *p* ≤ 0.05; ^†^ *p* ≤ 0.01; values represent Pearson’s correlation coefficients r; n = number of participants; MEEQ = Marijuana Expectancies of the Effects Questionnaire; LSAS = Liebowitz Social Anxiety Scale; CUPIT = Cannabis Use Problem Identification Test; F = female; CUD = Cannabis Use Disorder; MDD = Major Depressive Disorder; GAD = Generalized Anxiety Disorder; Educ. = highest levels of education; WC = White Caucasian race; EA = East Asian race; Occ. Status = occupational status; g = grams. ^1^ dummy coding for sex (0 = male, 1 = female); current CUD, Lifetime MDD, and current GAD (0 = no, 1 = yes); for highest level of education (0 = Less than university, 1 = university education or higher); race (W), (0 = other, 1 = White); race (EA), (0 = other, 1 = East Asian); income (0 = <$50,000, 1 = ≥$50,000), occupational status (0 = unemployed, 1 = employer or student); marital status (0 = single or separated, 1 = married or common law). ^2^ One drink is defined as either a 12-oz (341 mL) bottle of 5% alcohol beer or cider, a 5-oz (142 mL) glass of 12% alcohol wine, or a 1.5-oz (43 mL) shot glass of 40% alcohol spirits.

**Table 5 brainsci-14-00246-t005:** Simple bivariate correlation between clinical variables for participants with Social Anxiety Disorder (SAD).

(n = 26)	1	2	3	4	5	6	7	8	9	10	11	12	13	14	15
Total LSAS															
2. LSAS Anxiety	**0.95 ^†^**														
3. LSAS Avoidance	**0.97 ^†^**	**0.84 ^†^**													
4. Weekly cannabis use frequency	0.16	0.10	0.21												
5. The weekly amount of cannabis use (g)	0.28	0.24	0.30	**0.56 ^†^**											
6. Weekly alcohol use frequency (drinks ^1^/week)	0.01	−0.07	0.07	0.13	−0.25										
7. CUPIT	0.37	0.28	**0.42 ***	**0.77 ^†^**	**0.70 ^†^**	0.04									
8. MEEQ Total	0.28	0.36	0.19	−0.15	−0.22	−0.16	−0.04								
9. MEEQ Behavioral & Cognitive Impairment	0.01	−0.01	0.01	−0.30	−0.38	−0.04	−0.13	**0.72 ^†^**							
10. MEEQ Relaxation & Tension Reduction	**0.44 ***	**0.49 ***	0.37	0.23	0.30	−0.17	0.39	**0.42 ***	−0.10						
11. MEEQ Social & Sexual Facilitation	0.25	**0.40 ***	0.12	0.18	0.18	−0.17	0.03	**0.46 ***	−0.18	**0.56 ^†^**					
12. MEEQ Perceptual & Cognitive Enhancement	0.05	0.07	0.03	0.06	−0.22	−0.08	−0.06	**0.72 ^†^**	**0.47 ***	0.39	0.28				
13. MEEQ Global Negative Effects	0.19	0.18	0.18	−0.23	−0.12	0.18	−0.02	**0.48 ***	**0.56 ^†^**	−0.15	−0.20	0.22			
14. MEEQ Craving & Physical Effects	0.00	0.05	−0.04	**−0.51 ^†^**	**−0.53 ^†^**	−0.32	−0.36	**0.53 ^†^**	**0.62 ^†^**	−0.21	−0.03	0.15	0.22		
15. MEEQ Cumulative Negative Effects	0.07	0.06	0.08	−0.31	−0.32	0.04	−0.10	**0.72 ^†^**	**0.95 ^†^**	−0.13	−0.21	**0.43 ***	**0.79 ^†^**	**0.54 ^†^**	
16. MEEQ Cumulative Positive Effects	0.37	**0.50 ^†^**	0.25	0.06	0.07	−0.28	0.06	**0.63 ^†^**	0.03	**0.69 ^†^**	**0.88 ^†^**	**0.39 ***	−0.19	0.25	−0.05

* *p* ≤ 0.05, ^†^ *p* ≤ 0.01; values represent Pearson’s correlation coefficients r; n = number of participants; LSAS = Liebowitz Social Anxiety Scale; CUPIT = Cannabis Use Problems Identification Test; MEEQ = Marijuana Expectancies of the Effects Questionnaire; g = grams. ^1^ One drink is defined as either a 12-oz (341 mL) bottle of 5% alcohol beer or cider, a 5-oz (142 mL) glass of 12% alcohol wine, or a 1.5-oz (43 mL) shot glass of 40% alcohol spirits.

## Data Availability

The data are not publicly available due to their sensitive nature, and our ethical approval prevents us from sharing data beyond named collaborators. Further inquiries can be directed to the corresponding author B.L.F.

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
