# Peer review of "Expectancies of the Effects of Cannabis Use in Individuals with Social Anxiety Disorder (SAD)"

_brainsci, 2024, doi:10.3390/brainsci14030246_

Round 1
Reviewer 1 Report
Comments and Suggestions for Authors
The authors describe a study that aimed to examine differences between expectancies of the effects of cannabis use in adults with and without social anxiety disorder (SAD). Twenty-six individuals with and without SAD completed the Marijuana Expectancy and Effects Questionnaire. In a subsample, in-depth interviews were used to assess the trajectory of cannabis use perceptions. Both groups reported similar expectancies; however, in the SAD group, negative perceptions were reported before starting cannabis consumption, yet positive perceptions may have prompted initial use. The manuscript is well-written but would benefit from addressing the following:
Abstract:
1. Please briefly describe the analytical approach and the statistical outcomes (e.g., p-values, etc.).
2. The authors might want to expand their conclusions to discuss how the study contributes to the literature, given that cannabis perceptions and risks are typically part of cannabis use prevention approaches.
Introduction:
1. The introduction provides a clear rationale for the study and discusses the rigor of previous research.
2. It would be helpful if the authors provided scientific hypotheses in the last paragraph of the introductions.
3. What is the need to understand differences in expectancies/how they may change over time in individuals with social anxiety compared to those with diagnosed moderate-severe SAD?
Materials and Methods:
1. How many potential participants responded to recruitment efforts? How many completed pre-screening? How many were finally enrolled?
2. How many of the original participants were asked to participate in the in-depth interview to reach the target numbers?
3. Was other substance use assessed? Based on the literature, it would be important to assess other alcohol and substance use, as other use may influence findings.
4. How often were the participants in the substudy contacted? In other words, when were the 3 follow-up interviews conducted? Did all participants complete the 3 interviews? Were the interviews conducted telephonically or in person?
Discussion:
1. The limitations should include/discuss other substance use and co-use and how these behaviors influence expectancies and use patterns.
2. The authors may want to discuss how the study contributes to and advances the existing literature. Further, were the findings consistent with the hypotheses? If so, how? If not, what could explain this?
Author Response
Dear Reviewer 1,
Thank you so much for reviewing our article titled: “Expectancies of the Effects of Cannabis Use in Individuals with Social Anxiety Disorder (SAD)”. We have drafted our responses to your suggestions and requests below:
The authors describe a study that aimed to examine differences between expectancies of the effects of cannabis use in adults with and without social anxiety disorder (SAD). Twenty-six individuals with and without SAD completed the Marijuana Expectancy and Effects Questionnaire. In a subsample, in-depth interviews were used to assess the trajectory of cannabis use perceptions. Both groups reported similar expectancies; however, in the SAD group, negative perceptions were reported before starting cannabis consumption, yet positive perceptions may have prompted initial use. The manuscript is well-written but would benefit from addressing the following:
Abstract:
- Please briefly describe the analytical approach and the statistical outcomes (e.g., p-values, etc.).
We now have included some statistical values from our research in the abstract, however, we were unable to include all statistical information, given the 200-word limit. The following sentences were included in the abstract (lines 29 - 31, page 1) of the manuscript with tracked changes:
“While no between-group differences were observed, both groups reported expecting tension reduction and relaxation (F = 0.001; p=0.974), craving, and physical effects (F = 1.10; p = 0.300) but denied global negative effects (F = 0.11; p = 0.744).”
- The authors might want to expand their conclusions to discuss how the study contributes to the literature, given that cannabis perceptions and risks are typically part of cannabis use prevention approaches.
We now have changed our Abstract to include a discussion of our findings (lines 35 - 28, page 1; document with tracked changes), however, due to the 200-word limit, we were unable to expand on this topic further:
“Our data indicate that regardless of psychiatric history, frequent cannabis-using adults are more likely to report positive expectancies, associated with increased patterns of cannabis consumption. Psychoeducational programs and openly discussing the risks of cannabis may be beneficial in preventing and/or reducing cannabis use in SAD.”
Introduction:
- The introduction provides a clear rationale for the study and discusses the rigor of previous research.
- 2. It would be helpful if the authors provided scientific hypotheses in the last paragraph of the introductions.
We have now included our primary and secondary hypotheses in the Introduction (page 2, lines 147-149) that read:
“Based on previous research in emerging adults, we hypothesized that compared to their respective controls, adults with SAD will report more expectancies of cognitive and behavioral impairments..”
- What is the need to understand differences in expectancies/how they may change over time in individuals with social anxiety compared to those with diagnosed moderate-severe SAD?
Someone exhibiting social anxiety may not necessarily meet the criteria for SAD. Previous research was mostly focused on self-reported symptoms of social anxiety, which was not further assessed either by SCID-for DSM 5, nor assessed by healthcare workers qualified to determine psychiatric diagnosis. Thus, it is truly unknown if these individuals met the criteria for SAD per DSM-5 criteria. While there is some validity in assessing and treating only symptoms of disease or disorder, these symptoms may greatly vary among individuals, and they may or may not have a significant impact on functioning. The DSM-5 diagnosis ensures that a legitimate, standardized assessment is used in psychiatry, which by providing the criteria for diagnosis allows clinicians to establish commonalities among their patients. The establishment of these commonalities allows clinicians to better describe and communicate similarities, causes, and behaviors between their patients, and more importantly, apply treatments more adequately.
In our research, we were interested in assessing the expectancies of the effects of cannabis use in individuals who commonly had a certain group of symptoms diagnosable by SAD per DSM-5. By narrowing the psychiatric profile of our participants to focus exclusively on SAD, we hoped that we would be more efficient in determining the common expectancies, which would aid in determining the specific cannabis use patterns, and ultimately help clinicians plan drug prevention and harm reduction efforts. Moderate to severe SAD was selected because these individuals were more impacted by their symptoms.
Materials and Methods:
- How many potential participants responded to recruitment efforts? How many completed pre-screening? How many were finally enrolled?
329 potential participants (267 in the SAD and 62 in the control group) attended the pre-screening. 67/267 were invited to participate in the study in the SAD group and 32/62 in the control group. In the SAD group, 43/67 were consented (the rest did not show for their consenting interview), while 31/32 were consented in the control group. 17/43 participants in the SAD and 5/31 in the control group did not meet the screening criteria, leaving 26 participants in each group to complete the quantitative section of the study.
Please note that we now have revised our Study Procedures section (lines 186-211, page 3 in the document with tracked changes) to read:
“To all interested parties (329 participants; 267 for the SAD and 62 for the control group), a short pre-screening questionnaire was administered over the phone to provide a summary of the study procedures and conduct a preliminary screening for inclusion/exclusion criteria. Out of 329, 173 in the SAD group and 30 in the controls did not meet the preliminary criteria, while 27 in the SAD group were not interested in participating in the study. Interested study participants who met the preliminary study criteria (67 in the SAD and 32 in the control group) were invited to participate in the study. Before attending their test day, verbal informed consent was obtained over the phone. Out of 67 invited participants in the SAD group, 43 consented, while 27 were no-shows at the consenting phone interviews. In the control group out of 32 enrolled, 31 consented, whereas only 1 participant did not attend the consenting process. The consent to participate in the study was documented in the Informed Consent Form (ICF) explicitly designed for online participation. Subsequently, participants attended one online WebEx session on the test day. The screening session included administering SCID-5 RV [28] to determine if the participants met eligibility criteria, followed by the screening checklist for personality disorders, LSAS [27], Medical and Psychiatric History Form, and Sociodemographic Assessment Form. In the SAD group 17/43 did not meet the study criteria during screening, while 5/31 failed screening in the control group, Those who met the eligibility criteria (26 in the SAD and 26 in the control group) participated in the study session, which included administering the Marijuana Expectancy of the Effects Questionnaire [19,20] the Cannabis Use Problems Identification Test (CUPIT) [29], and the Drug/Alcohol Use History Form. The first twelve participants from each group were invited to an in-depth interview, on a first come first serve basis. All invited participants accepted to attend in-depth interviews, except one male participant with SAD because he felt uncomfortable answering open-ended questions. Participation in in-depth interviews was optional. All data for this study was collected using paper-based tools.”
- 2. How many of the original participants were asked to participate in the in-depth interview to reach the target numbers?
12 participants from each group were asked on a first come first serve basis to participate in in-depth interviews. Only one male subject with SAD declined.
We now have revised our Study Procedures section to clarify the recruitment of participants in in-depth interviews (lines 207 - 211, page 3 in the document with tracked changes) as follows:
“The first twelve participants from each group were invited to an in-depth interview, on a first come first serve basis. All invited participants accepted to attend in-depth interviews, except one male participant with SAD because he felt uncomfortable answering open-ended questions. Participation in in-depth interviews was optional. All data for this study was collected using paper-based tools.”
- Was other substance use assessed? Based on the literature, it would be important to assess other alcohol and substance use, as other use may influence findings.
Please, note that the participants in both groups with any other substance use disorders, including alcohol use disorder were excluded from the study, as our focus was on cannabis use expectancies, rather than on the effects of polysubstance use on cannabis use expectancies.
However, we have collected information on other substances and alcohol use by administering Drug/Alcohol Use History Form. We have now included some information about the form in our manuscript in the Methods Section, (page 3, lines 203 - 207, and page 4, lines 230- 236 in the document with tracked changes) which read:
“Those who met the eligibility criteria (26 in the SAD and 26 in the control group) participated in the study session, which included administering the Marijuana Expectancy of the Effects Questionnaire [19,20] the Cannabis Use Problems Identification Test (CUPIT) [29], and the Drug/Alcohol Use History Form.”
“Drug/Alcohol Use History Form captured a frequency of lifetime and current (in the past month) alcohol and drug use. Besides, cannabis, alcohol, and cigarette use, other types of substance use recorded in this form included sedatives (benzodiazepines, barbiturates, sleeping pills, and quaaludes), stimulants (methamphetamines, amphetamines, ADHD medications, and other diet pills), cocaine, opioids (heroin, morphine, oxycodone, opium, methadone, and Fentanyl), dissociative anesthetics (PCP, ketamine, GHB), and hallucinogens (LSD, mushrooms, MDMA and peyote).”
Moreover, Table 1 was revised to include the means and standard deviations of weekly frequency of alcohol use (in terms of the number of drinks consumed) in both groups and the statistical comparison of the means between the groups. In the footnote, Table 1 further describes a definition of a drink:
“One drink is defined as either a 12-oz (341 ml) bottle of 5% alcohol beer or cider, a 5-oz (142 ml) glass of 12% alcohol wine, or a 1.5-oz (43 ml) shot glass of 40% alcohol spirits.”
We now have included weekly alcohol consumption in bivariate correlational analyses in Table 4, in which this parameter is compared to various demographic characteristics in the SAD group. In Table 5, weekly alcohol consumption is correlated to clinical variables, as well as various MEEQ domains in the SAD group. However, no statistically significant correlations were observed between weekly alcohol consumption and any demographic or clinical variables.
Also, in the Results section, we have included information on drug use (pages 5-6, lines 311-316 and lines 330 – 351 in the document with tracked changes), which reads:
“Although participants with co-morbid substance and alcohol use disorders were excluded from the study, no significant differences between groups were observed when comparing the alcohol use frequency. Moreover, our analysis of the use of other recreational substances showed that 2/26 (7.7%) of SAD and 2/26 (7.7%) in the control group used sedatives (χ2 = 0; p = 1.0). Of those four, three reported using them no more than 10x per month, while only one participant in the SAD reported using benzodiazepines in the last month, but at a frequency of no more than 10 times. The stimulant use was observed in 3/26 (11.5%) in SAD and 4/26 (15.4%) (χ2 = 0.17; p = 0.685) in the control group, which was mostly consumed no more than 10x/month in the past. Of those, one participant in the control group admitted to using crystal methamphetamine in the past month, however, he used it no more than 10x. The lifetime cocaine use was reported in 8/26 (30.8%) participants in SAD and 6/26 (23.1%) in the control group (χ2 = 0.39 p = 0.532). Of the 14, 13 participants did not use it currently (in the past month) or more than 10x/month at any time in the past. One participant from the SAD group, however, admitted to using cocaine in the past month, but no more than 10x. Only one participant from each group (3.8% in SAD and 3.8% in the control group; χ2 = 0; p = 1.0) indicated using opioids, but neither one of them was using currently, nor 10x/month in the past. For dissociative anesthetics, there were no current reports of the use, although 2/26 (7.7%) in the SAD and 1/26 (3.8%) in the control group reported (χ2 = 0.35 p = 0.552) the lifetime use of no more than 10x.month. Hallucinogens, specifically mushrooms were the most popular illicit substance among both groups. In the SAD group, 12/26 (46.2%) and 10/26 (38.5%) of controls (χ2 = 0.32 p = 0.575) reported lifetime use, of which 3/26 (11.5%) of SAD and 2/26 (7.7%) used hallucinogens in the past month, but no more than 10x. Only 3 participants in the SAD and 4 in the control group were regular cigarette users, who met the criteria for Nicotine Use Disorder. Considering that the use of substances listed above was overall infrequent and the current use minimal, their impact on the expectancies of the effects of cannabis use was not assessed, except for alcohol.”
As mentioned in the text above, given that a very small percentage of participants used other substances at the time of the assessment, no further analysis of the impact of these substances was conducted in the study. Tables 4 and 5 did not show any correlation between alcohol consumption and outcome measures. Also, the impact of the use of other substances was not revealed during in-depth interviews, which assessed the trajectory of cannabis use perceptions.
- How often were the participants in the substudy contacted? In other words, when were the 3 follow-up interviews conducted? Did all participants complete the 3 interviews? Were the interviews conducted telephonically or in person?
All three sections were conducted during one online (via Webex) study session. As indicated below, in-depth interviews were administered to 12/26 participants from each group on a first-come, first-serve basis. The study assessments were conducted using Webex technology.
Please, note that this information is included in the Study Procedures section, which reads (page 3, lines 198 – 211 in the document with tracked changes):
“Subsequently, participants attended one online WebEx session on the test day. The screening session included administering SCID-5 RV [28] to determine if the participants met eligibility criteria, followed by the screening checklist for personality disorders, LSAS [27], Medical and Psychiatric History Form, and Sociodemographic Assessment Form. In the SAD group 17/43 did not meet the study criteria during screening, while 5/31 failed screening in the control group, Those who met the eligibility criteria (26 in the SAD and 26 in the control group) participated in the study session, which included administering the Marijuana Expectancy of the Effects Questionnaire [19,20] the Cannabis Use Problems Identification Test (CUPIT) [29], and the Drug/Alcohol Use History Form. The first twelve participants from each group were invited to an in-depth interview, on a first come first serve basis. All invited participants accepted to attend in-depth interviews, except one male participant with SAD because he felt uncomfortable answering open-ended questions. Participation in in-depth interviews was optional. All data for this study was collected using paper-based tools.”
Discussion:
- The limitations should include/discuss other substance use and co-use and how these behaviors influence expectancies and use patterns.
We have now revised our Limitations section to include the following (page 18, lines 829 – 837 in the document with tracked changes):
“Our study focused on examining the expectancies of the effects of cannabis in individuals with and without SAD, who did not meet the criteria for alcohol use disorder, or any other substance use disorders. Although we demonstrated no correlation between alcohol use frequency and demographics, clinical characteristics of our participants, and MEEQ outcomes, future studies may assess the contribution of other substance use disorders or alcohol dependence on the expectancies of the effects of cannabis use in SAD. The consideration of performing such an assessment is due to the reported polysubstance use in individuals with SAD [65, 66].”
- The authors may want to discuss how the study contributes to and advances the existing literature. Further, were the findings consistent with the hypotheses? If so, how? If not, what could explain this?
The information describing the consistency of our study with the hypothesis, as well as the explanation of our findings related to the hypothesis are stated in the first paragraph of the discussion, which reads (pages 14-15, lines 621-634 in the document with tracked changes) as follows:
“The primary objective of our study was to compare expectancies of the effects of cannabis use between individuals with and without SAD. Our data showed no significant differences across all six lower-order and two higher-order MEEQ questionnaire domains between the two groups. Contrary to the work conducted by Buckner and colleagues, [9,10] MEEQ assessment indicated that individuals with SAD did not have higher expectancies of global negative effects or cognitive-behavioral impairments with cannabis use. Moreover, the average scores for both groups demonstrated higher expectations of relaxation and tension reduction, and craving and physical effects, as well as higher-order positive effects. Our findings were consistent with the expectancy theory [18] and previous studies [19,20] indicating that regardless of their psychiatric history, frequent cannabis users, such as those recruited in our study, were more likely to report positive and less likely negative expectancies of cannabis use. Accordingly, these positive expectancies may have driven repeated cannabis consumption, regardless of the experienced consequences, such as developing cannabis use addiction or dependence.”
Moreover, in our Discussion section based on our regression analysis, we attempt to explain the results from the previous research on the expectancies of the effects of cannabis use in emerging adults, in the following paragraphs (page 15, lines 635-648 in the document with tracked changes) as follows:
“Our regression analysis revealed that age was negatively correlated to cognitive and behavioral impairment and global negative effects. Moreover, we showed that within the SAD group, age was negatively correlated to cognitive negative impairment and higher-order negative effects. Our finding may explain the results produced by the previous studies, showing occurrent negative expectancies in emerging adults with SAD with narrow age groups [9,10]. High school students and young college students with SAD are often exposed to unique social settings of their vocational environments in which they are constantly surrounded by groups of people. Such settings may be social anxiety-triggering [2-5], prompting these individuals to feel like they are benefiting from cannabis-related impairments [9,10]. They may use cannabis as a ‘camouflage’ for their self-perceived inadequate social behavior, which they may attribute to cannabis impairment rather than their personality [9,10,18]. However, adults with SAD (like participants from our study) are no longer exposed to such social situations, as they are no longer in school, and may no longer need to use cannabis to cover up their self-perceived behavior. “
Thank you again for reviewing our manuscript. Please do not hesitate to let us know should you need further clarification regarding our responses.
Sincere regards,
Sonja Elsaid on behalf of all authors
Reviewer 2 Report
Comments and Suggestions for Authors
Thanks for this practical research.
1- Although the number of samples is statically acceptable, More samples would increase the value of the research.
2-The number of references below in 2018 is significant.
3-Overall,it is a valuable research.
Author Response
Dear Reviewer 2,
Thank you so much for reviewing our article titled: “Expectancies of the Effects of Cannabis Use in Individuals with Social Anxiety Disorder (SAD)”. We have drafted our responses to your suggestions and requests below:
Thanks for this practical research.
1- Although the number of samples is statically acceptable, More samples would increase the value of the research.
We agree with this statement and recognize that a larger sample size would allow a more elaborate regression analysis, as indicated in our Limitations section (page 18, lines 837-841 in the document with tracked changes), which reads:
“Lastly, we recruited 26 participants per group, which allowed adequate comparison of expectancies of the effects of cannabis use at a cannabis use maintenance stage. Although our sample size was sufficient to assess the impact of major covariates such as age, sex, and current CUD, a larger sample size would have allowed us to examine the broader effects of demographic and clinical variables on the expectancy data. “
2-The number of references below in 2018 is significant.
Most of the previous research on this topic was conducted before 2018, therefore many of our references listed are from papers published before 2018, except a few review articles summarizing the topic.
- Single, A.; Bilevicius, E.; Ho, V.; Theule, J.; Buckner, J. D.; Mota, N.; Keough, M. T., Cannabis use and social anxiety in young adulthood: A meta-analysis. Addictive behaviors 2022, 107275.
- Buckner, J. D.; Morris, P. E.; Abarno, C. N.; Glover, N. I.; Lewis, E. M., Biopsychosocial model social anxiety and substance use revised. Current Psychiatry Reports 2021, 23, 1-9.
3-Overall,it is a valuable research.
Thank you.
Thank you again for reviewing our manuscript. Please do not hesitate to let us know should you need further clarification regarding our responses.
Sincere regards,
Sonja Elsaid on behalf of all authors
Reviewer 3 Report
Comments and Suggestions for Authors
General comments:
1. The paper presents an empirical study on expectancies of the effects of cannabis use in individuals with Social Anxiety Disorder.
2. The manuscript is well-written and understandable.
3. The research topic is of relevance both for research and practice.
4. The paper could nicely fit into the Special Issue Addictive and Concomitant Psychiatric Disorders.
Specific comments:
5. There may be a major difference between expectancies about the drug use effects and the actual motivation to use drugs. Unless we do not know the actual motivations, we cannot say if Social Anxiety Disorder is linked to that or not. Thus, the conclusions have to remain somewhat speculative.
6. The sample size is relatively small. Are estimates reliable with such a small sample? Has an a priori power analysis been conducted? Were there outliers?
7. The possibility to address education policies is raised. However, the sample comprises adults, not children / adolescents. A strong educative focus may already target early life phases.
8. The role of the social network in increasing motivation to use drugs can be discussed in more depth.
9. The practical relevance could be highlighted.
10. What about those individuals that claim to use drugs because of medical reasons (i.e., like a treatment). What about other reasons?
Author Response
Dear Reviewer 3,
Thank you so much for reviewing our article titled: “Expectancies of the Effects of Cannabis Use in Individuals with Social Anxiety Disorder (SAD)”. We have drafted our responses to your suggestions and requests below:
The paper presents an empirical study on expectancies of the effects of cannabis use in individuals with Social Anxiety Disorder.
- The manuscript is well-written and understandable.
- The research topic is of relevance both for research and practice.
- The paper could nicely fit into the Special Issue Addictive and Concomitant Psychiatric Disorders.
Specific comments:
- There may be a major difference between expectancies about the drug use effects and the actual motivation to use drugs. Unless we do not know the actual motivations, we cannot say if Social Anxiety Disorder is linked to that or not. Thus, the conclusions have to remain somewhat speculative.
Please, note that we have already published our data on motivations for cannabis use in the same population sample in 14. Elsaid, S.; Wang, R.; Kloiber, S.; Le Foll, B.; Hassan, A. N., Motivations for Cannabis Use in Individuals with Social Anxiety Disorder (SAD). Brain Sciences 2023, 13 (12), 1698 doi: 10.3390/brainsci13121698., thus none of our statements related to the motivations of cannabis use in this population sample are speculative but based on our previous research.
For instance, in the following sentences in our Discussion section, we drew the parallel between motivations and expectancies of the effects of cannabis use by referencing our previous work (page 16, lines 707-711 in the document with tracked changes):
“The top three positive views included enhancement, coping and relaxation, and social facilitation. Participants with SAD also reported expecting that cannabis assisted them as a coping mechanism for their SAD, possibly prompting much earlier consumption than their respective controls together with their higher motivations for coping and enhancement [14].”
Similarly in the Conclusion section, we have referred to our previous research in the following sentences (page 19, lines 859-861):
“The acknowledgment of expecting tension reduction and relaxation specifically among SAD participants aligned with our previous study on the same cohort, demonstrating increased cannabis use motivations for coping.”
- The sample size is relatively small. Are estimates reliable with such a small sample? Has an a priori power analysis been conducted? Were there outliers?
Priori sample size calculations were conducted before initiating the study. The specific information on the steps used is now included in the Data analysis section (pages 4-5, lines 262-270 in the document with tracked changes) as follows:
“Considering that previous research mainly focused on emerging adults and not on the general adult populations, our sample size calculations were conducted on the data presented in [8] from emerging adults, 18 – 22 years of age (mean = 19.13 years of age, standard deviation = 1.07) [8]. G*Power 3.1 (Heinrich Heine University of Dusseldorf, Germany) was used for computing the effects and sample size calculations based on the means and standard deviations of the MEEQ Cognitive and Behavioral Impairments domain (hypothesis), used from SAD and non-SAD participants [8]. The derived effect size was Cohen’s d = 0.82, which was entered with 1 - β = 0.8 and α = 0.05 to determine that 26 participants per group were a sufficient sample size for our quantitative analysis.
For qualitative research, the sample size of 12 participants per group was sufficient, considering that the saturation point related to the themes discussed was reached with this sample size. We have discussed the sample size considerations in our Data Analysis section (page 5, lines 297-300), as follows:
“The saturation point related to the themes discussed during the interviews was reached with 12 participants per group, as no new, additional information was shared while interviewing our last group of participants. Henceforth, this sample size was quite sufficient for our study.”
There were very few outliers, 1 in the control group for the Cognitive and Behavioral Impairment, 1 in the SAD group for Enhancement, 2 in the control for Global Negative Effects, and 1 for the SAD group for Global Negative Effects, 0 for Tension and Reduction and 0 for Social and Sexual Facilitation domains. The outliers were kept as they represented the true values of participants in each group (i.e. they were not entered in error).
- The possibility to address education policies is raised. However, the sample comprises adults, not children / adolescents. A strong educative focus may already target early life phases.
We now have added some information about the potential preventive and harm-reduction strategies that could be considered for adults with SAD (page 18, lines 796-816 in the document with tracked changes) as follows:
Specific prevention and harm reduction efforts could also be implemented to benefit adults with SAD. For instance, primary healthcare providers should engage in discussing cannabis use with non-using clients with SAD or suspected SAD. The discussion could acknowledge the perceived anecdotal benefits of cannabis on symptoms of SAD while balancing the explanation of perceived immediate versus long-term benefits and harms (such as cannabis addiction) [62]. Effects of cannabis and its components on polypharmacy and drug-drug interaction could be assessed, specifically in older adults and those affected by multiple physical and/or other psychiatric conditions. Cognitively impaired individuals should be aware of additional adverse effects they may experience due to cannabis consumption, should they choose to do so [62,63]. The harm reduction strategies for those with SAD could focus on reducing the negative consequences associated with cannabis use. For instance, discouraging cannabis use in situations where its consumption may pose a risk (e.g. while driving a car, caring for a child, before work, or in public places) or using small amounts of cannabis gradually and seeing its effects to prevent significant intoxication could also be considered [62]. Prompting individuals with SAD to pay more attention to the types of ratios of cannabinoids on product labels could also be beneficial, especially given that it was previously shown that higher THC in cannabis was linked to anxiogenic effects in those already suffering from anxiety-related symptoms [62,64]. Most importantly, clinicians and mental health workers could specifically be made aware to watch for symptoms of SAD in adolescents and adults to encourage rapid diagnosis and the administration of conventional treatments for SAD.
Please, note that we specifically referred to the Preventure program that has never been considered (to our knowledge) in adolescents with a diagnosis or the potential diagnosis of SAD by the following (page 17, lines 775-785 in the document with tracked changes):
“For example, one such program tested in controlled randomized trials is the Preventure program, designed to target adolescents at risk of engaging in alcohol and substance use [59-61]. The risk assessment is based on identifying personality traits (e.g. hopelessness, anxiety sensitivity, impulsivity, and sensation seeing). Once identified, the high-risk individuals receive interventions held during school hours, which include personality-tailored psycho-educational, motivational enhancement, and cognitive behavioral therapy. The data from clinical trials indicates that the Preventure program was effective in reducing alcohol and substance use harm by 50% [59-61]. Specifically useful could be similar programs designed to identify SAD in schoolchildren younger than 16 years of age when they are at risk of starting cannabis use.”
- The role of the social network in increasing motivation to use drugs can be discussed in more depth.
Please, note that our paper does not focus on examining the motivations for cannabis use, but the expectancies of the effects of cannabis use in SAD. Moreover, we have addressed the role of the social network in increasing motivations to use cannabis in this cohort in our previous paper: “Elsaid, S.; Wang, R.; Kloiber, S.; Le Foll, B.; Hassan, A. N., Motivations for Cannabis Use in Individuals with Social Anxiety Disorder (SAD). Brain Sciences 2023, 13 (12), 1698 doi: 10.3390/brainsci13121698.
Nevertheless, the role of social networks on the expectancies of the effects of cannabis use is addressed in the Discussion section (page 15, lines 635-648 in the document with tracked changes), as follows:
“Our regression analysis revealed that age was negatively correlated to cognitive and behavioral impairment and global negative effects. Moreover, we showed that within the SAD group, age was negatively correlated to cognitive negative impairment and higher-order negative effects. Our finding may explain the results produced by the previous studies, showing occurrent negative expectancies in emerging adults with SAD with narrow age groups [9,10]. High school students and young college students with SAD are often exposed to unique social settings of their vocational environments in which they are constantly surrounded by groups of people. Such settings may be social anxiety-triggering [2-5], prompting these individuals to feel like they are benefiting from cannabis-related impairments [9,10]. They may use cannabis as a ‘camouflage’ for their self-perceived inadequate social behavior, which they may attribute to cannabis impairment rather than their personality [9,10,18]. However, these unique adults are no longer exposed to such social situations, as they are no longer in school, and may no longer need to use cannabis to cover up their self-perceived behavior.”
- The practical relevance could be highlighted.
The practical relevance of our research was highlighted in the Introduction section (page 2, lines 115-120 in the document with tracked changes) as follows:
“To better plan treatment efforts, clinicians have been looking for ways to predict different patterns of cannabis use. One promising area that addresses the topic is the expectancy of the effects of drug use theory [18], which also applies to cannabis use. The theory posits that drug-related perceptions may influence drug consumption behavior. Accordingly, expecting positive effects was linked to greater frequency of cannabis use, whereas expecting adverse effects were negatively correlated [19-21].”
- 10. What about those individuals that claim to use drugs because of medical reasons (i.e., like a treatment). What about other reasons?
Please, note that in the SAD group individuals with serious medical conditions or psychiatric disorders and those suffering from acute or chronic pain were excluded from the study, thus eliminating the possibility that cannabis was used for reasons other than recreationally. Moreover, only individuals whose chief complaint was SAD (in the SAD group), and who did not use cannabis for any other comorbidity were included in the study. In the control group, any individuals with medical or psychiatric conditions were also excluded from the study. Also, in the SAD group, no individuals reported being prescribed cannabis for SAD by their healthcare workers. The purpose of such inclusion/exclusion criteria was to focus on recreational cannabis use in individuals with and without SAD.
Thank you again for reviewing our manuscript. Please do not hesitate to let us know should you need further clarification regarding our responses.
Sincere regards,
Sonja Elsaid on behalf of all authors